# Deep Learning for Black-Box Modeling of Audio Effects †

**Marco A. Martínez Ramírez \*, Emmanouil Benetos**  **and Joshua D. Reiss**

Centre for Digital Music, Queen Mary University of London, Mile End Road, London E1 4NS, UK;
emmanouil.benetos@qmul.ac.uk (E.B.); joshua.reiss@qmul.ac.uk (J.D.R.)

\* Correspondence: m.a.martinezramirez@qmul.ac.uk

† This paper is an extended version of our paper published in the International Conference on Digital Audio
Effects (DAFx-19), Birmingham, UK, 4–7 September 2019.

**Abstract:** Virtual analog modeling of audio effects consists of emulating the sound of an audio processor reference device. This digital simulation is normally done by designing mathematical models of these systems. It is often difficult because it seeks to accurately model all components within the effect unit, which usually contains various nonlinearities and time-varying components. Most existing methods for audio effects modeling are either simplified or optimized to a very specific circuit or type of audio effect and cannot be efficiently translated to other types of audio effects. Recently, deep neural networks have been explored as black-box modeling strategies to solve this task, i.e., by using only input–output measurements. We analyse different state-of-the-art deep learning models based on convolutional and recurrent neural networks, feedforward WaveNet architectures and we also introduce a new model based on the combination of the aforementioned models. Through objective perceptual-based metrics and subjective listening tests we explore the performance of these models when modeling various analog audio effects. Thus, we show virtual analog models of nonlinear effects, such as a tube preamplifier; nonlinear effects with memory, such as a transistor-based limiter and nonlinear time-varying effects, such as the rotating horn and rotating woofer of a Leslie speaker cabinet.

**Keywords:** black-box modeling; nonlinear; time-varying; audio effects; deep learning; tube amplifier; transistor-based limiter; Leslie speaker

## 1. Introduction

Modeling of virtual analog audio effects is the process of emulating an audio effect unit and seeks to recreate the sound, behaviour and main perceptual features of an analog reference device [1]. Audio effect units are analog or digital signal processing systems that transform certain characteristics of the sound source. These transformations can be linear or nonlinear, time-invariant or time-varying and with short-term and long-term memory. Most typical audio effect transformations are based on dynamics, such as compression, tone such as distortion, frequency such as equalization, and time such as artificial reverberation or modulation based audio effects.

Nonlinear audio effects: These effects are widely used by musicians and sound engineers and can be classified into two main types of effects: dynamic processors such as compressors or limiters; and distortion effects such as tube amplifiers [2]. Distortion effects are mainly used for aesthetic reasons and are usually applied to electric musical instruments such as electric guitar, bass guitar, electric piano or synthesizers. The main sonic characteristic of these effects is due to their nonlinearities and the most common processors are overdrive, distortion pedals, tube amplifiers and guitar pickup emulators.

Dynamic range processors are nonlinear time-invariant audio effects with long temporal dependencies, and their main purpose is to alter the variation in volume of the incoming audio. This is achieved with a varying amplification gain factor, which depends on an envelope follower along with a waveshaping nonlinearity. These effects tend to introduce a low amount of harmonic distortion, while for tube amplifiers a strong distortion is desired [2].

Thus, distortion effects and dynamic range processors are based on the alteration of the waveform which leads to various degrees of amplitude and harmonic distortion. The nonlinear behavior of certain components of the effects' circuit performs this alteration, which can be seen as a waveshaping nonlinearity applied to the amplitude of the incoming audio signal in order to add harmonic and inharmonic overtones. For example, a waveshaping transformation depends on the amplitude of the input signal and consists in using a nonlinear function, such as an hyperbolic tangent, to distort the shape of the incoming waveform [3].

Modulation based audio effects: Modulation based or time-varying audio effects involve audio processors that include a modulator signal within their analog or digital implementation [4]. These modulator signals are in the low frequency range (usually below 20 Hz). Their waveforms are based on common periodic signals such as sinusoidal, squarewave or sawtooth oscillators and are often referred to as a Low Frequency Oscillator (LFO). The LFO periodically modulates certain parameters of the audio processors altering the timbre, frequency, loudness or spatialization characteristics of the audio. Based on how the LFO is employed and the underlying signal processing techniques used when designing the effect units, we can classify modulation based audio effects into *time-varying filters* such as phaser or wah-wah; *delay-line based* effects such as flanger or chorus; and *amplitude modulation* effects such as tremolo or ring modulator [2].

The Leslie speaker cabinet is a type of modulation based effect that combines amplitude, frequency and spatial modulation. It consists of a vacuum-tube amplifier and crossover filter followed by a rotating *horn* and rotating *woofer* inside a wooden cabinet. This effect can be interpreted as a combination of tremolo, Doppler effect and reverberation [5].

Audio effects modeling: Modeling these types of effect units or analog circuits has been heavily researched and remains an active field, see Section 2 for more details. Virtual analog methods for modeling nonlinear and time-varying audio effects mainly involve circuit modeling and optimization for specific analog components such as vacuum-tubes, operational amplifiers or transistors. This often requires models that are too specific for a certain circuit or making certain assumptions when modeling specific nonlinearities. Therefore such models are not easily transferable to different effects units since expert knowledge of the type of circuit being modeled is always required. Also, musicians tend to prefer analog counterparts because their digital implementations may lack the broad behaviour of the analog reference devices.

Recently, deep learning architectures have been explored for black-box modeling of audio effects. In previous works, we explored convolutional neural networks (CNN) to model linear effect units, such as equalization [6]; nonlinear effects with short-term memory, such as distortion, overdrive and amplifier emulation [7]. Furthermore, in [8], the later architecture was extended with recurrent neural networks (RNN) in order to model linear and nonlinear, time-varying and time-invariant audio effects with long temporal dependencies, such ring modulation or multiband compression. Also, in [9], Damskägg et al. explored variants of the WaveNet architecture [10] in order to model nonlinear effects such as a tube amplifier.

In this work, we analyse and compare the deep learning architectures from [7–9] and we propose a new model based on the combination of the convolutional and dense architectures from [8] with the feedforward WaveNet from [9]. Therefore, we explore whether a latent-space based on WaveNet can learn long temporal dependencies such as those learned by the Bidirectional Long-Short Term Memory (Bi-LSTM) layers from [8].

We show the models performing virtual analog modeling of the *Universal Audio vacuum-tube preamplifier 610-B*, the *Universal Audio transistor-based limiter amplifier 1176LN* and the rotating *horn*

and rotating *woofer* of a *145 Leslie speaker cabinet*. We measure the performance of the models via perceptually-based objective metrics and through a subjective listening test. We report that convolutional and feedforward WaveNet architectures perform similarly when modeling nonlinear audio effects without memory and with long temporal dependencies, but fail to model time-varying tasks such as the *Leslie speaker*. On the other hand, and across all tasks, the models that incorporate RNNs or WaveNet architectures to explicitly learn long temporal dependencies, tend to outperform (objectively and subjectively) the rest of the models.

The paper is structured as follows. In Section 2 we present the relevant literature related to modeling nonlinear and time-varying audio effects and Table 1 summarizes the different approaches. Section 3 provides the description of the different deep learning models and Section 4 the experimental procedures. Sections 5–7, respectively show the obtained results, discussion and conclusions.

**Table 1.** Summary of approaches for virtual analog modeling of audio effects.

| Type | Audio Effect | Approach | | Reference |
|---|---|---|---|---|
| nonlinear with short-term memory | tube amplifier | static waveshaping | | [11] |
| | tube amplifier | dynamic nonlinear filters | | [12] |
| | distortion | static waveshaping & numerical methods | | [13] |
| | distortion | circuit simulation | K-method & WDF | [14] |
| | distortion | circuit simulation | Nodal DK | [15] |
| | speaker, amplifier | analytical method | Volterra series | [16] |
| | Moog ladder filter | analytical method | Volterra series | [17] |
| | power amplifier | black-box | Wiener & Hammerstein | [18] |
| | distortion | black-box | Wiener | [19] |
| | tube amplifer | black-box | Wiener-Hammerstein | [20] |
| | equalization | black-box | end-to-end DNN | [6] |
| | tube amplifier | black-box | end-to-end DNN | [21] |
| | tube amplifier | black-box | end-to-end DNN | [22] |
| | equalization & distortion | black-box | end-to-end DNN | [7] |
| | tube amplifier | black-box | end-to-end DNN | [9] |
| | tube amplifier, distortion | black-box | end-to-end DNN | [23] |
| | distortion | circuit simulation & DNN | | [24] |
| time-dependent nonlinear | compressor | circuit simulation | state-space | [25] |
| | compressor | black-box | system-identification | [26] |
| | compressor | gray-box | system-identification | [27] |
| | compressor | gray-box | end-to-end DNN | [28] |
| time-varying | ring modulator | static waveshaping | | [29] |
| | phaser | circuit simulation | numerical methods | [30] |
| | phaser | circuit simulation | Nodal DK | [31] |
| | modulation based with OTAs | circuit simulation | WDF | [32] |
| | flanger with BBDs | circuit simulation | Nodal DK | [33] |
| | modulation based with BBDs | circuit simulation & system-identification | | [32] |
| | Leslie speaker horn | digital filter-based & system identification | | [34] |
| | Leslie speaker horn & woofer | digital filter-based | | [35] |
| | Leslie speaker horn & woofer | digital filter-based | | [36] |
| | flanger, chorus | digital filter-based | | [30] |
| | modulation based with BBDs | digital filter-based | | [37] |
| | modulation based | gray-box | system-identification | [38] |
| | modulation based & compressor | black-box | end-to-end DNN | [8] |

## 2. Background

### 2.1. Modeling of Nonlinear Audio Effects

Since a nonlinear system cannot be characterized by its impulse response, frequency response or transfer function [1], digital emulation of distortion effects have been extensively researched [39]. Different methods have been proposed such as *memoryless static waveshaping* [11], where system-identification methods are used to approximate the nonlinearity; *dynamic nonlinear filters* [12], where the waveshaping curve changes its shape as a function of the input signal or system-state variables; *circuit simulation* techniques [13–15], where a complete study of the analog circuitry is performed and nonlinear filters are derived from the differential equations that describe the circuit; and *analytical methods* [16,17], where the nonlinearity is modeled via Volterra series theory or nonlinear black-box approaches such as Wiener and Hammerstein models [18–20].

Modeling of dynamic range processors, such as compressors, has been based on white-box methods such as *circuit simulation*, where a complete study of the internal circuit is carried out; and black-box methods such as *system identification* techniques, where a model is structured using only the measurements of the input and output signals. In [25], state-space models are used to simulate the circuit of an specific analog guitar compressor. Black-box [26] and gray-box [27] modeling of general-purpose dynamic range compressors has been investigated via input–output measurements and optimization routines. The latter differs from black-box modeling, since gray-box approaches use some information about the circuit together with input–output signals.

Generalization among different audio effect units is usually difficult since these methods are often either simplified or optimized to a very specific circuit. This lack of generalization is accentuated when we consider that each audio processor is also composed of components other than the nonlinearity. These components also need to be modeled and often involve filtering before and after the nonlinearity, as well as short and long temporal dependencies such as hysteresis or attack and release gates.

### 2.2. Modeling of Time-Varying Audio Effects

Most research for modeling time-varying audio effects has been explored via white-box methods. In order to model the various analog components that characterize the circuitry of this type of effects, circuit simulation approaches are based on diodes [29], transistors [30,31], operational transconductance amplifiers (OTAs) [32] or integrated circuits such as Bucket Brigade Delay (BBD) chips [33,37,40]. Common methods for circuit simulation include the nodal DK-method [41] and Wave Digital Filters (WDF) [42]. By assuming linear behaviour or by omitting certain nonlinear circuit components, most of these effects can be implemented directly in the digital domain through the use of digital filters and delay lines. In [38], based on all-pass filters and multiple measurements of impulse responses, a gray-box modeling method for linear time-varying audio effects is proposed.

The *Leslie speaker* cabinet represents a special case of modulation based audio effects, since amplitude and frequency modulation occur along with the reverberation and structural resonance of the wooden cabinet. In [34], the rotating *horn* of the *Leslie speaker* is modeled via varying delay-lines, artificial reverberation and physical measurements from the rotating loudspeaker. Likewise, [35,36] modeled the *Leslie speaker horn* and *woofer* through time-varying spectral delay filters and time-varying FIR filters, respectively. In these *Leslie speaker* emulations, various physical characteristics of the effect are not taken into account, such as the frequency-dependent directivity of the loudspeakers or the effect of the wooden cabinet.

### 2.3. Deep Learning for Audio Effects Modeling

Deep learning architectures for audio processing tasks, such as audio effects modeling, have been investigated as end-to-end methods or as parameter estimators of audio processors [43,44]. End-to-end deep learning architectures, where raw audio is both the input and the output of the system, follow

black-box modeling approaches where an entire problem can be taken as a single indivisible task which must be learned from input to output. The desired output is obtained by learning and processing directly from the incoming raw audio, thus reducing the amount of required prior knowledge and minimizing the engineering effort [45].

End-to-end deep neural networks (DNNs) for audio effects modeling have been recently explored for linear and nonlinear, time-varying and time-invariant audio effects with long temporal dependencies. Equalization matching is achieved in [6] and nonlinear modeling in [7], where the network is capable of modeling an arbitrary combination of linear and nonlinear audio effects with short-term memory. Nevertheless, the network of [7] does not generalize to transformations with long temporal dependencies such as modulation based audio effects. The model is divided into three parts: adaptive front-end, latent-space and synthesis back-end, and follows an adaptive convolutional architecture together with dense layers and trainable activation functions as nonlinear waveshapers.

Several linear and nonlinear time-varying and time-invariant audio effects were modeled in [8], following the adaptive convolutional architecture from [7]. The structure of the synthesis back-end is modified and RNNs are incorporated into the latent-space in order to explore their capabilities when learning transformations with long temporal dependencies.

Also, based on [46], a feedforward variant of the WaveNet architecture is proposed in [9], where a nonlinear audio effect and its controls are emulated. This network outperforms current state-of-the-art analytical methods for nonlinear black-box modeling such as the block-oriented Wiener models presented in [19].

In [28], gray-box modeling is proposed for nonlinear effects with long temporal dependencies such as compressors. The architecture is based on U-Net [47] and Time-Frequency [48] networks, where using input–output measurements and knowledge of the attack and release gate times are used to emulate different compressors and their respective controls. Similarly, RNNs for real-time black-box modeling of tube amplifiers and distortion pedals were explored in [23] and static configurations of tube amplifiers in [21,22]. A gray-box method is explored in [24], where a DNN is used to model the state-space system of nonlinear distortion circuits.

## 3. Methods

In this section we present the architecture of the different black-box audio effects modeling networks: the deep convolutional audio effects modeling architecture (*CAFx*) from [7], the feedforward *WaveNet* from [9] and the convolutional and recurrent audio effects modeling architecture (*CRAFx*) from [8]. Also, we introduce *CWAFx*, a combination of the convolutional, dense and activation layers from *CRAFx* together with a latent-space based WaveNet. All the models are based entirely in the time-domain and end-to-end; with raw audio as the input and processed audio as the output. Code is availabe online (https://mchijmma.github.io/DL-AFx/). Also, Appendix A shows the number of parameters and processing times across all models.

### 3.1. Convolutional Audio Effects Modeling Network: CAFx

The model is divided into three parts: adaptive front-end, synthesis back-end and latent-space DNN. The architecture is designed to model nonlinear audio effects with short-term memory and is based on a parallel combination of cascade input filters, trainable waveshaping nonlinearities, and output filters. All convolutions are along the time dimension and all strides are of unit value. This means, during convolution, we move the filters one sample at a time. The model is depicted in Figure 1 and its structure is described in detail in Table 2. We use an input frame of size 4096 sampled with a hop size of 2048 samples.

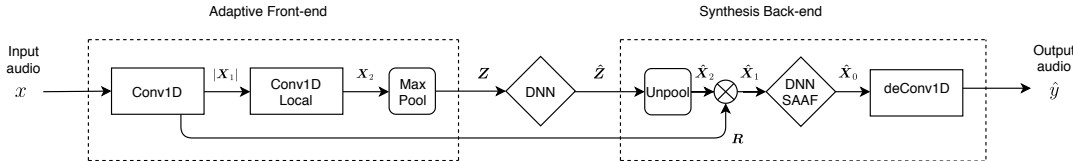

**Figure 1.** Block diagram of *CAFx*; adaptive front-end, synthesis back-end and latent-space DNN.

**Table 2.** Detailed architecture of *CAFx* with an input frame size of 4096 samples.

| Layer | Output Shape | Weights | Output |
|---|---|---|---|
| Input | (4096, 1) | . | $x$ |
| Conv1D | (4096, 128) | 128(64) | $X_1$ |
| Residual | (4096, 128) | . | $R$ |
| Abs | (4096, 128) | . | . |
| Conv1D-Local | (4096, 128) | 128(128) | $X_2$ |
| MaxPooling | (64, 128) | . | $Z$ |
| Dense-Local | (128, 64) | 64(128) | . |
| Dense | (128, 64) | 64 | $\hat{Z}$ |
| Unpooling | (4096, 128) | . | $\hat{X}_2$ |
| $R \times \hat{X}_2$ | (4096, 128) | . | $\hat{X}_1$ |
| Dense | (4096, 128) | 128 | . |
| Dense | (4096, 64) | 64 | . |
| Dense | (4096, 64) | 64 | . |
| Dense | (4096, 128) | 128 | . |
| SAAF | (4096, 128) | 128(25) | $\hat{X}_0$ |
| deConv1D | (4096, 1) | . | $\hat{y}$ |

The **adaptive front-end** consists of a convolutional encoder. It contains two CNN layers, one pooling layer and one residual connection. The first convolutional layer is followed by the *absolute value* as nonlinear activation function and the second convolutional layer is locally connected. This means that we follow a filterbank architecture since each filter is only applied to its corresponding row in the input feature map. This layer is followed by the *softplus* nonlinearity. The *max-pooling* layer is a moving window of size 64, where the maximum value within each window corresponds to the output and the positions of the maximum values are stored and used by the back-end. The operation performed by the first layer can be described by (1):

$$X_1 = x * W_1 \tag{1}$$

where $*$ denotes the convolution operator, $W_1$ is the kernel matrix from the first layer, and $X_1$ is the feature map after the input audio $x$ is convolved with $W_1$. The weights $W_1$ consist of 128 one-dimensional filters of size 64. The residual connection $R$ is equal to $X_1$, which corresponds to the frequency band decomposition of the input $x$.

The operation performed by the second layer is described by (2):

$$X_2 = softplus(|X_1| * W_2) \tag{2}$$

where $X_2$ is the second feature map obtained after the locally connected convolution with $W_2$, the kernel matrix of the second layer which has 128 filters of size 128.

The adaptive front-end performs time-domain convolutions with the raw audio and is designed to learn a latent representation for each audio effect modeling task. It also generates a residual connection which is used by the back-end to facilitate the synthesis of the waveform based on the specific audio effect transformation. By using the *absolute value* as activation function of the first layer and by having larger filters $W_2$, we expect the front-end to learn smoother representations of the incoming audio, such as envelopes [49].

The **latent-space DNN** contains two layers. Following the filter bank architecture, the first layer is based on locally connected dense layers and the second layer consists of a fully connected (FC) layer. The DNN modifies the latent representation $Z$ into a new latent representation $\hat{Z}$ which is fed into the synthesis back-end. The first layer applies a different dense layer to each row of the matrix $Z$ and the second layer is applied to each row of the output matrix from the first layer. In both layers, all dense layers have 64 hidden units, are followed by the *softplus* function and are applied to the complete latent representation rather than to the channel dimension.

The **synthesis back-end** accomplishes the nonlinear task by the following steps. First, $\hat{X}_2$, the discrete approximation of $X_2$, is obtained via unpooling the modified envelopes $\hat{Z}$. Then the feature map $\hat{X}_1$ is the result of the element-wise multiplication of the residual connection $R$ and $\hat{X}_2$. This can be seen as an input filtering operation, since a different envelope gain is applied to each of the frequency band decompositions obtained in the front-end.

The second step is to apply various waveshapping nonlinearities to $\hat{X}_1$. This is achieved with a a DNN with smooth adaptive activation functions (DNN-SAAF). The DNN-SAAF consists of 4 FC dense layers. All dense layers are followed by the *softplus* function with the exception of the last layer. Locally connected Smooth Adaptive Activation Functions (SAAFs) [50] are used as the nonlinearity for the last layer. SAAFs consist of piecewise second order polynomials which can approximate any continuous function and are regularized under a Lipschitz constant to ensure smoothness. Overall, each function is locally connected and composed of 25 intervals between $-1$ to 1.

We tested different standard and adaptive activation functions, such as the parametric and non parametric rectifier linear unit (*ReLU*), hyperbolic tangent, sigmoid and fifth order polynomials. Nevertheless, we found stability problems and non optimal results when modeling nonlinear effects. Since each SAAF explicitly acts as a waveshaper, the DNN-SAAF is constrained to behave as a set of trainable waveshaping nonlinearities, which follow the filter bank architecture and are applied to the channel dimension of the modified frequency decomposition $\hat{X}_1$.

Finally, the last layer corresponds to the deconvolution operation, which can be implemented by transposing the first layer transform. This layer is not trainable since its kernels are transposed versions of $W_1$. In this way, the back-end reconstructs the audio waveform in the same manner that the front-end decomposed it. The complete waveform is synthesized using a *hann* window and constant overlap-add gain.

### 3.2. Feedforward WaveNet Audio Effects Modeling Network—WaveNet

The *WaveNet* architecture corresponds to a feedforward variation of the original autoregressive model. For a regression task, such as nonlinear modeling, the predicted samples are not fed back into the model, but through a sliding input window, where the model predicts a set of samples in a single forward propagation. The feedforward Wavenet implementation is based on the architecture proposed in [9] and [46]. The model is divided into two parts: stack of dilated convolutions and a post-processing block. The model is depicted in Figure 2 and its structure is described in Table 3.

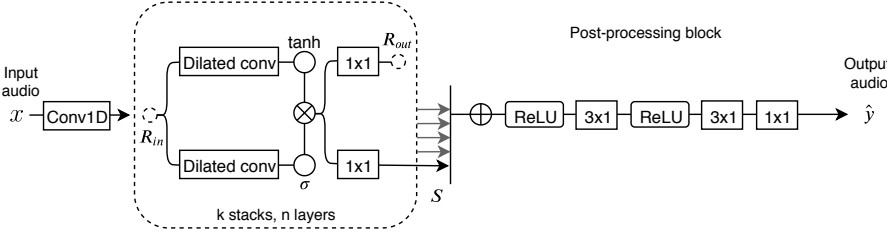

**Figure 2.** Block diagram of the feedforward *WaveNet*; stack of dilated convolutional layers and the post-processing block.

**Table 3.** Detailed architecture of *WaveNet* with input and output frame sizes of 5118 and 4096 samples respectively.

| Layer–Output Shape–Weights | | Output | |
|---|---|---|---|
| Input (5118, 1) | | $x$ | |
| Conv1D (5118, 16)–16(3) | | $R_{in}$ | |
| Dilated conv (5118, 16)–16(3) | Dilated conv (5118, 16)–16(3) | . | |
| Tanh (5118, 16) | Sigmoid (5118, 16) | . | . |
| Multiply (5118, 16) | | $z$ | |
| Conv1D (5118, 16)–16(1) | Conv1D (5118, 16)– 16(1) | $R_{out}$ | $S$ |
| Add (4096, 16) | | . | |
| ReLU (4096, 16) | | . | |
| Conv1D (4096, 2048)–2048(3) | | . | |
| ReLU (4096, 16) | | . | |
| Conv1D (4096, 256)–256(3) | | . | |
| Conv1D (4096, 1)–1(1) | | $\hat{y}$ | |

We use 2 stacks of 8 dilated convolutional layers with a dilation factor of $1, 2, \ldots, 128$ and 16 filters of size of 3. From Figure 1, prior to the stack of dilated convolutions, the input $x$ is projected into 16 channels via a $3 \times 1$ convolution. This in order to match the number of channels within the feature maps of the dilated convolutions.

The **stack of dilated convolutions** processes the input feature map $R_{in}$ with $3 \times 1$ gated convolutions and exponentially increasing dilation factors. This operation can be described by:

$$z = tanh(\boldsymbol{W}_f * \boldsymbol{R}_{in}) \cdot \sigma(\boldsymbol{W}_g * \boldsymbol{R}_{in}) \tag{3}$$

Where $\boldsymbol{W}_f$ and $\boldsymbol{W}_g$ are the filter and gated convolutional kernels, *tanh* and $\sigma$ the hyperbolic tangent and sigmoid functions and $*$ and $\cdot$ the operators for convolution and element-wise multiplication. The residual output connection $\boldsymbol{R}_{out}$ and the skip conection $S$ are obtained via a $1 \times 1$ convolution applied to $z$. Thus, $S$ is sent to the post-processing block and $\boldsymbol{R}_{out}$ is added to the current input matrix $R_{in}$, thus, resulting in the residual input feature map of the next dilated convolutional layer.

The **post-processing block** consists in summing all the skip connections $S$ followed by a *ReLU*. Two final $3 \times 1$ convolutions are applied to the resulting feature map, which contain 2048 and 256 filters and are separated by a *ReLU*. As a last step, a $1 \times 1$ convolution is introduced in order to obtain the single-channel output audio $\hat{y}$.

Since the receptive field of the model is of 1022 samples, in order to output frames of 4096 samples, the input presented to the model consists of sliding frames of 5118 samples.

### 3.3. Convolutional Recurrent Audio Effects Modeling Network—CRAFx

The *CRAFx* model builds on the *CAFX* architecture and is also divided into three parts: adaptive front-end, latent-space and synthesis back-end. A block diagram can be seen in Figure 3 and its structure is described in detail in Table 4. The main difference is the incorporation of Bi-LSTMs into the latent-space and the modification of the synthesis back-end structure. This in order to allow the model to learn nonlinear transformations with long temporal dependencies. Also, instead of 128 channels, due to the training time of the recurrent layers, this model uses 32 channels.

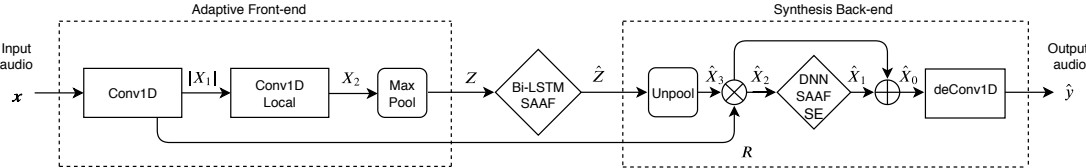

**Figure 3.** Block diagram of *CRAFx*; adaptive front-end, latent-space Bi-LSTM and synthesis back-end.

**Table 4.** Detailed architecture of a model with input frame size of 4096 samples and $\pm 4$ context frames.

| Layer | Output Shape | Weights | Output |
|---|---|---|---|
| Input | (9, 4096, 1) | . | $x$ |
| Conv1D | (9, 4096, 32) | 32(64) | $X_1$ |
| Residual | (4096, 32) | . | $R$ |
| Abs | (9, 4096, 32) | . | . |
| Conv1D-Local | (9, 4096, 32) | 32(128) | $X_2$ |
| MaxPooling | (9, 64, 32) | . | $Z$ |
| Bi-LSTM | (64, 128) | 2(64) | . |
| Bi-LSTM | (64, 64) | 2(32) | . |
| Bi-LSTM | (64, 32) | 2(16) | . |
| SAAF | (64, 32) | 32(25) | $\hat{Z}$ |
| Unpooling | (4096, 32) | . | $\hat{X}_3$ |
| Multiply | (4096, 32) | . | $\hat{X}_2$ |
| Dense | (4096, 32) | 32 | . |
| Dense | (4096, 16) | 16 | . |
| Dense | (4096, 16) | 16 | . |
| Dense | (4096, 32) | 32 | . |
| SAAF | (4096, 32) | 32(25) | $\hat{X}'_1$ |
| Abs | (4096, 32) | . | . |
| Global Average | (1, 32) | . | . |
| Dense | (1, 512) | 512 | . |
| Dense | (1, 32) | 32 | $se$ |
| $\hat{X}'_1 \times se$ | (4096, 32) | . | $\hat{X}_1$ |
| $\hat{X}_1 + \hat{X}_2$ | (4096, 32) | . | $\hat{X}_0$ |
| deConv1D | (4096, 1) | . | $\hat{y}$ |

In order to allow the model to learn long-term memory dependencies, the input consists of the current audio frame $x$ concatenated with the 4 previous and 4 subsequent frames. These frames are of size 4096 and sampled with a hop size $\tau = 2048$ samples. The input $x$ is described by:

$$x = x(t + j\tau), j = -4, ..., 4 \tag{4}$$

The **adaptive front-end** is exactly the same as the one from *CAFx*, but its layers are time distributed, i.e., the same convolution or pooling operation is applied to each of the 9 input frames. In this model, $R$ is the corresponding row in $X_1$ for the frequency band decomposition of the current input frame $x$. Thus, the back-end does not directly receive information from the past and subsequent context frames.

The **latent-space** consists of three Bi-LSTM layers of 64, 32, and 16 units, respectively. Bi-LSTMs are a type of RNN that can access long-term context from both backward and forward directions [51]. Bi-LSTMs are capable of learning long temporal dependencies when processing time series where the context of the input is needed [52].

The Bi-LSTMs process the latent-space representation $Z$, which is learned by the front-end and contains information regarding the 9 input frames. These recurrent layers are trained to reduce the dimension of $Z$, while also learning the modulators $\hat{Z}$. This new latent representation is fed into the synthesis back-end in order to reconstruct an audio signal that matches the modeling task. Each Bi-LSTM has dropout and recurrent dropout rates of 0.1 and the first two layers have *tanh* as activation function. Also, the nonlinearities of the last recurrent layer are locally connected SAAFs.

The **synthesis back-end** accomplishes the reconstruction of the target audio by processing the frequency band decomposition $R$ and the nonlinear modulation $\hat{Z}$. The new structure of the back-end incorporates a Squeeze-and-Excitation (SE) [53] layer after the DNN-SAAF block (DNN-SAAF-SE).

The SE block explicitly models interdependencies between channels by adaptively scaling the channel-wise information of feature maps [53]. Thus, we propose a SE block which applies a dynamic gain to each of the feature map channels and follows the structure from [54]. It consists of a global average pooling operation followed by two FC layers. The FC layers are followed by *ReLU* and *sigmoid*

activation functions accordingly. Since the feature maps of the model are based on time-domain waveforms, we incorporate an *absolute value* layer before the global average pooling operation.

Following the filter bank architecture, the back-end matches the time-varying task by the following steps. First, an upsampling operation is applied to the learned modulators $\hat{Z}$ which is followed by an element-wise multiplication with the residual connection $R$. This can be seen as a frequency dependent amplitude modulation to each of the channels or frequency bands of $R$. This is followed by the nonlinear waveshaping and channel-wise scaled filters from the DNN-SAAF-SE block.

Thus, the modulated frequency band decomposition $\hat{X}_2$ is processed by the learned waveshapers from the DNN-SAAF layers and further scaled by the frequency dependent gains from the SE layers. The resulting feature map $\hat{X}_1$ can be seen as modeling the nonlinear short-term memory transformations within the audio effects modelling tasks. Then, $\hat{X}_1$ is added back to $\hat{X}_2$, acting as a nonlinear feedforward delay line. The structure of the back-end is informed by the general architecture in which the modulation based effects are implemented in the digital domain, through the use of LFOs, digital filters and delay lines.

Finally, the complete waveform is synthesized in the same way as in *CAFx*, where the last layer corresponds to the transposed and non-trainable deconvolution operation.

### 3.4. Convolutional and WaveNet Audio Effects Modeling Network - CWAFx

We propose a new model based on the combination of the convolutional and dense architectures from *CRAFx* with the dilated convolutions from *WaveNet*. Since the Bi-LSTM layers in the former were in charge of learning long temporal dependencies from the input and context audio frames, we replace these recurrent layers with a feedforward WaveNet. As it has been shown that dilated convolutions outperform recurrent approaches when learning sequential problems [55], such as in [56], where Bi-LSTMs are successfully replaced with this type of temporal convolutions.

Thus, we investigate whether a latent-space based on stacked dilated convolutions can learn frequency-dependent amplitude modulation signals. The model is depicted in Figure 4 and the structure of the **latent-space WaveNet** is described in detail in Table 5. The **adaptive front-end** and **synthesis back-end** are the same as the ones presented in *CRAFx*.

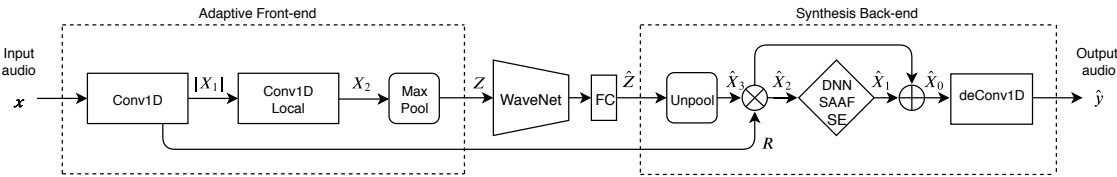

**Figure 4.** Block diagram of *CWAFx*; adaptive front-end, latent-space WaveNet and synthesis back-end.

**Table 5.** Detailed architecture of the latent-space WaveNet.

| Layer–Output Shape–Weights | | Output | |
|---|---|---|---|
| $Z$ (576, 32) | | . | |
| Conv1D (576, 32)–32(3) | | $R_{in}$ | |
| Dilated conv (576, 32)–32(3) | Dilated conv (576, 32) –32(3) | . | |
| Tanh (576, 32) | Sigmoid (576, 32) | . | . |
| Multiply (576, 32) | | . | |
| Conv1D (576, 32)–32(1) | Conv1D (576, 32)–32(1) | $R_{out}$ | $S$ |
| Add (576, 32) | | . | |
| ReLU (576, 32) | | . | |
| Conv1D (576, 32)–32(3) | | . | |
| ReLU (576, 32) | | . | |
| Conv1D (576, 32)–32(3) | | . | |
| Dense (32, 64)–64 | | $\hat{Z}$ | |

The latent representation $Z$ from the front-end corresponds to 9 rows of 64 samples and 32 channels, which can be unrolled into a feature map of 576 samples and 32 channels. Thus, we approximate these input dimensions with a latent-space WaveNet with receptive and target fields of 510 and 64 samples, respectively. We use 2 stacks of 7 dilated convolutional layers with a dilation factor of 1, 2, . . . , 64 and 32 filters of size 3. Also, we achieved better fitting by keeping the dimensions of the skip connections $S$ and by replacing the final $1 \times 1$ convolution with a FC layer. The latter has 64 hidden units followed by the *tanh* activation function and is applied along the latent dimension.

## 4. Experiments

### 4.1. Training

The training of the *CAFX*, *CRAFx* and *CWAFx* architectures includes an initialization step. This pretraining stage consists in optimizing a network formed solely by the convolutional and pooling layers of the front-end and back-end. This pretraining allows to have a better fitting when training for the nonlinear or time-varying tasks. Thus, within an unsupervised learning task, this network is trained to process and reconstruct both the dry audio *x* and target audio *y*. Only during this step the unpooling layer of the back-end uses the time positions of the maximum values recorded by the *max-pooling* operation.

Once the front-end and back-end are pretrained, the rest of the convolutional, recurrent, dense and activation layers are incorporated into the respective models, and all the weights are trained following an end-to-end supervised learning task. The *WaveNet* model is trained directly following this second step. Since small amplitude errors are as important as large ones, the loss function to be minimized is the mean absolute error between the target and output waveforms.

For both training steps, *Adam* [57] is used as optimizer and we use an early stopping patience of 25 epochs, i.e., training stops if there is no improvement in the validation loss. The model is fine-tuned further with the learning rate reduced by a factor of 4 and also a patience of 25 epochs. The initial learning rate is $1 \times 10^{-4}$ and the batch size consists of the total number of frames per audio sample. On average, the total number of epochs is approximately 750. We select the model with the lowest error for the validation subset (see Section 4.2). For the *Leslie speaker* modeling tasks, the early stopping and model selection procedures were based on the training loss. This is explained in more detail in Section 6.

### 4.2. Dataset

Raw recordings of individual 2-second notes of various 6-string electric guitars and 4-string bass guitars are obtained from the *IDMT-SMT-Audio-Effects* dataset [58]. We use the 1250 unprocessed recordings of electric guitar and bass to obtain the wet samples of the respective audio effects modeling tasks. The raw recordings are amplitude normalized and for each task the test and validation samples correspond to 5% of this dataset each. After the analog audio processors were sampled with the raw notes, all the recordings were downsampled to 16 kHz. The dataset is available online (https://mchijmma.github.io/DL-AFx/).

#### 4.2.1. Universal Audio Vacuum-Tube Preamplifier 610-B

This microphone tube preamplifier (*preamp*) is sampled from a *6176 Vintage Channel Strip* unit. In order to obtain an output signal with high harmonic distortion, the *preamp* is overdriven with the following settings: gain +10 dB, level 6, line impedance and high and low boost/cut 0 dB.

4.2.2. Universal Audio Transistor-Based Limiter Amplifier 1176LN

Similarly, the wildly used field-effect transistor *limiter 1176LN* is sampled from the same *6176 Vintage Channel Strip* unit. The *limiter* samples are recorded with the following settings: attack 800 µs, release 1100 ms, input level 4, output level 7 and ratio *ALL*. We use the slowest attack and release settings in order to further test the long-term memory of the models. The compression ratio value of *ALL* corresponds to all the ratio buttons of an original 1176 being pushed simultaneously. Thus, this setting also introduces distortion due to the variation of attack and release times.

4.2.3. 145 Leslie Speaker Cabinet

The output samples from the rotating *horn* and *woofer* of a *145 Leslie speaker* cabinet are recorded with a *AKG-C451-B* microphone. Each recording is done in mono by placing the condenser microphone perpendicularly to the *horn* or *woofer* and 1 meter away. Two speeds are recorded for each rotating speaker; *tremolo* for a fast rotation and *chorale* for a slow rotation. The rotation frequency of the *horn* is approximately 7 Hz and 0.8 Hz for the *tremolo* and *chorale* settings, respectively, while the *woofer* has slower speed rotations [36].

Since the *horn* and *woofer* speakers are preceded by a 800 Hz crossover filter, we apply a highpass FIR filter with the same cutoff frequency to the raw notes of the electric guitar and use only these samples as input for the *horn* speaker. Likewise, for the *woofer* speaker we use a lowpass FIR filter to preprocess the raw bass notes. The audio output of both speakers is filtered with the respective FIR filters. This in order to reduce mechanical and electrical noise and also to focus the modeling tasks on the amplitude and frequency modulations. Also, the recordings are amplitude-normalized.

*4.3. Objective Metrics*

Three metrics are used when testing the models with the various modeling tasks. Since the mean absolute error depends on the amplitude of the output and target waveforms, before calculating this metric, we normalize the energy of the target and the output and define it as the energy-normalized mean absolute error (*mae*).

As an objective evaluation for the *Leslie speaker* time-varying tasks, we propose an objective metric which mimics human perception of amplitude and frequency modulation. The modulation spectrum uses time-frequency theory integrated with the psychoacoustics of modulation frequency perception, thus, providing long-term knowledge of temporal fluctuation patterns [59]. The modulation spectrum mean squared error (*ms_mse*) is based on the audio features from [60] and [61] and is defined as follows:

- A Gammatone filter bank is applied to the target and output entire waveforms. In total we use 12 filters, with center frequencies spaced logarithmically from 26 Hz to 6950 Hz.
- The envelope of each filter output is calculated via the magnitude of the Hilbert transform and downsampled to 400 Hz.
- A Modulation filter bank is applied to each envelope. In total we use 12 filters, with center frequencies spaced logarithmically from 0.5 Hz to 100 Hz.
- The Fast Fourier Transform (FFT) is calculated for each modulation filter output of each Gammatone filter. The energy is summed across the Gammatone and Modulation filter banks and the *ms_mse* metric is the mean squared error of the logarithmic values of the FFT frequency bins.

The evaluation for the nonlinear tasks with short-term and long-term memory corresponds to *mfcc_cosine*: the mean cosine distance of the Mel-frequency cepstral coefficients (MFCCs). This metric is calculated as follows:

- A log-power-melspectogram is computed from the energy-normalized waveforms. This is calculated with 40 mel-bands and audio frames of 4096 samples and 50% hop size.
- 13 MFCCs are computed using the discrete cosine transform and the *mfcc_cosine* metric is the mean cosine distance across the MFCC vectors.

*4.4. Listening Test*

Thirty participants between the ages of 23 and 46 took part in the experiment which was conducted at a professional listening room at Queen Mary University of London. The Web Audio Evaluation Tool [62] was used to set up the test and participants used *Beyerdynamic DT-770 PRO* studio headphones.

The subjects were among musicians, sound engineers or experienced in critical listening. The listening samples were obtained from the test subsets and each page of the test contained a reference sound, i.e., a recording from the original analog device. The aim of the test was to identify which sound is closer to the reference, and participants rated 6 different samples according to the similarity of these in relation to the reference sound.

Therefore, participants were informed what modeling task they were listening to, and were asked to rate the samples from 'least similar' to 'most similar'. This in a scale of 0 to 100, which was then mapped into a scale of 0 to 1. The samples consisted of a dry sample as anchor, outputs from the 4 different models and a hidden copy of the reference.

## 5. Results

The training procedures were performed for each architecture and each modeling task: *preamp* corresponds to the vacuum-tube preamplifier, *limiter* to the transistor-based limiter amplifier, *horn tremolo* and *horn chorale* to the *Leslie speaker* rotating horn at fast and slow speeds, respectively, and *woofer tremolo* and *woofer chorale* to the rotating woofer at the corresponding speeds. Then, the models were tested with samples from the test subset and the audio results are available online (https://mchijmma.github.io/DL-AFx/).

Figure 5 shows the *mae*, *mfcc_cosine* and *ms_mse* for all the test subsets. It can be seen that the *mae* models' performance is similar within each modeling tasks with *limiter* having the lowest error. Also, *CAFx* presents the largest errors, with the *Leslie speaker chorale* settings being the highest.

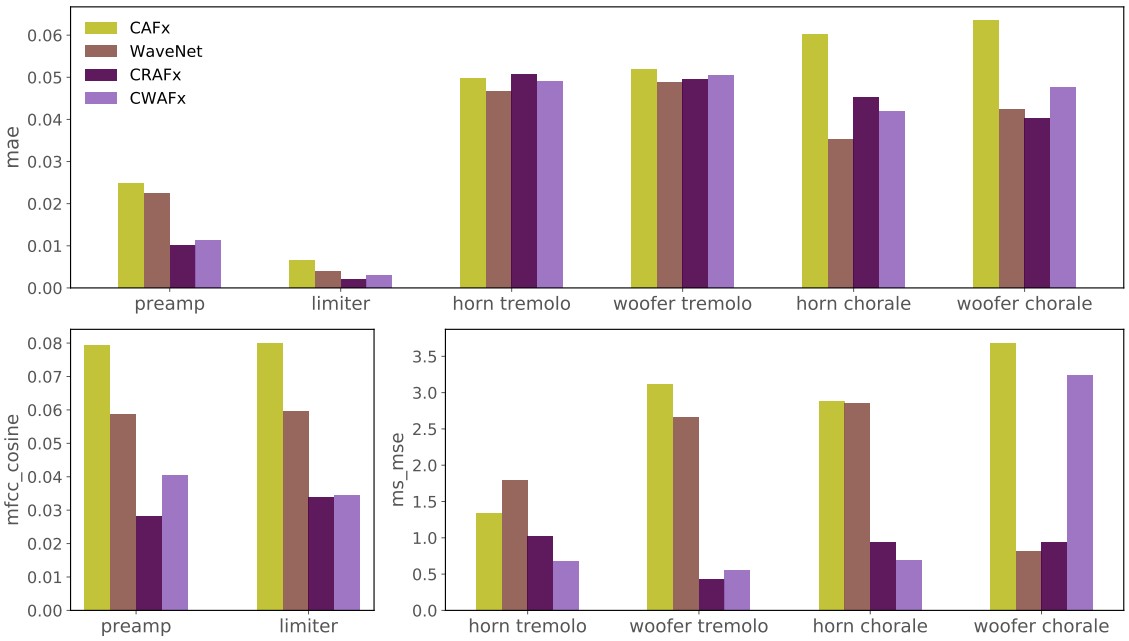

**Figure 5. mae**, **mfcc_cosine** and **ms_mse** values with the test dataset for all the modeling tasks.

In terms of perceptually-based metrics such as the *mfcc_cosine* and *ms_mse*, the *CRAFx* and *CWAFx* models achieved the best scores. This with the exception of the *woofer chorale* task, where the *CWAFx* model did not manage to accomplish the task. Overall, *CRAFx* and *CAFx* correspond to the highest and lowest scoring models, respectively.

The results of the listening test for all modeling tasks can be seen in Figure 6 as notched box plots. The end of the notches represents a 95% confidence interval and the end of the boxes represent the first and third quartiles. Also, the green lines illustrate the median rating and the purple circles represent outliers. In general, both anchors and hidden references have the lowest and highest median, respectively. The perceptual findings match closely the objective metrics from Figure 5, since the architectures that explicitly learn long-temporal dependencies, such as *CRAFx* and *CWAFx* outperform the rest of the models. Furthermore, for the *woofer chorale* task, the unsuccessful performance of the latter is also evidenced in perceptual ratings. This indicates that the latent-space WaveNet fails to learn low-frequency modulations such as the *woofer chorale* rotating rate.

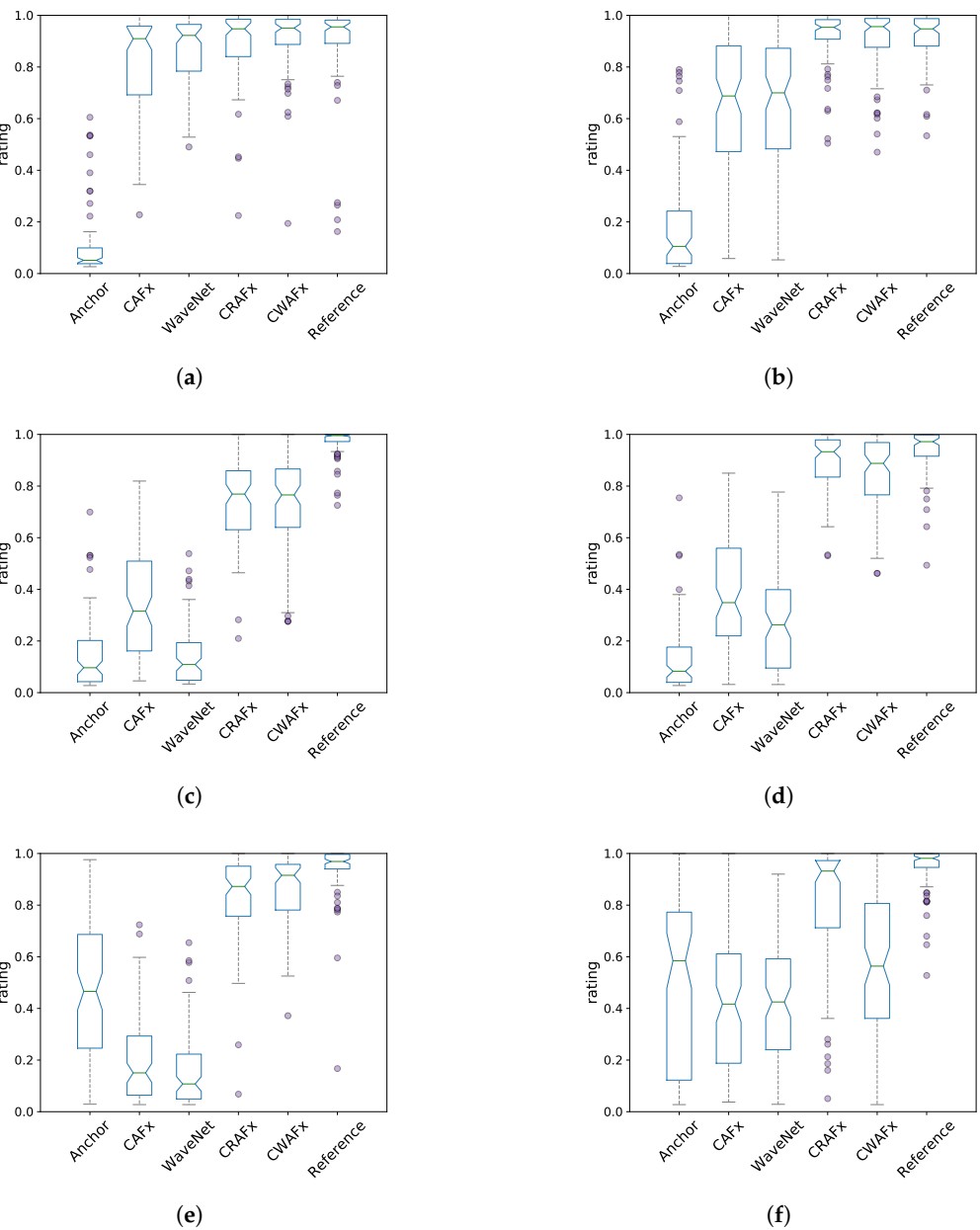

**Figure 6.** Box plot showing the rating results of the listening tests. (**a**) **preamp**, (**b**) **limiter**, (**c**) **Leslie speaker horn-tremolo**, (**d**) **Leslie speaker woofer-tremolo**, (**e**) **Leslie speaker horn-chorale** and (**f**) **Leslie speaker woofer-chorale**.

For selected test samples of the *preamp* and *limiter* tasks and for all the different models, Figure 7 shows the input, reference, and output waveforms together with their respective spectrogram. Both in the time-domain and in the frequency-domain, it is observable that the waveforms and spectrograms are in line with the objective and subjective findings. To more closely display the performance of these nonlinear tasks, Figure 8 shows a segment of the respective waveforms. It can be seen how the different models match the waveshaping from the overdriven *preamp* as well as the attack waveshaping of the *limiter* when processing the onset of the test sample.

Regarding the *Leslie speaker* modeling task, Figures 9–12 show the different waveforms together with their respective modulation spectrum and spectrogram: Figure 9 *horn-tremolo*, Figure 10 *woofer-tremolo*, Figure 11 *horn-chorale* and Figure 12 *woofer-chorale*. From the spectra, it is noticeable that *CRAFx* and *CWAFx* introduce and match the amplitude and frequency modulations of the reference, whereas *CAFX* and *WaveNet* fail to accomplish the time-varying tasks.

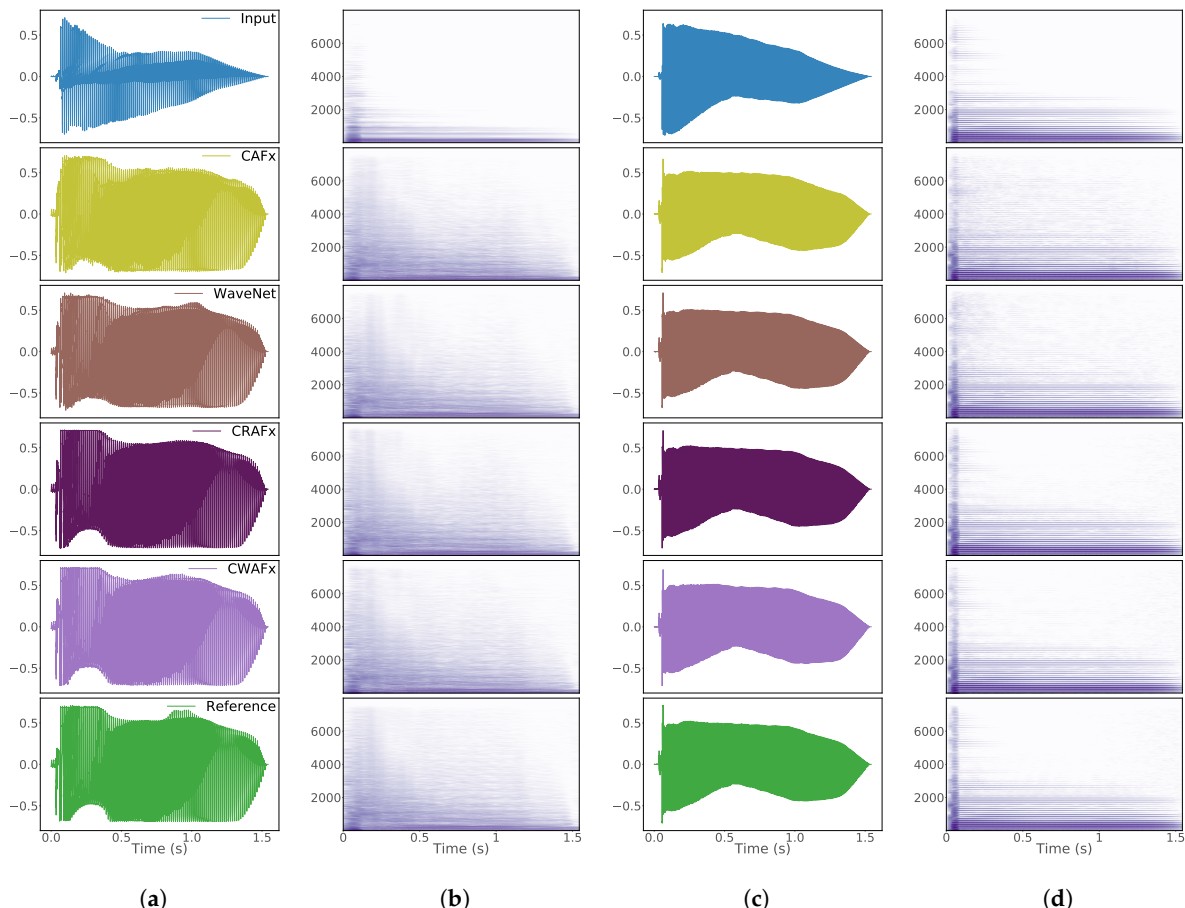

**Figure 7.** Results with selected samples from the test dataset for the tasks: (**a**,**b**) **preamp** and (**c**,**d**) **limiter**. The waveforms and their respective spectrograms are shown and vertical axes represent amplitude and frequency (Hz), respectively.

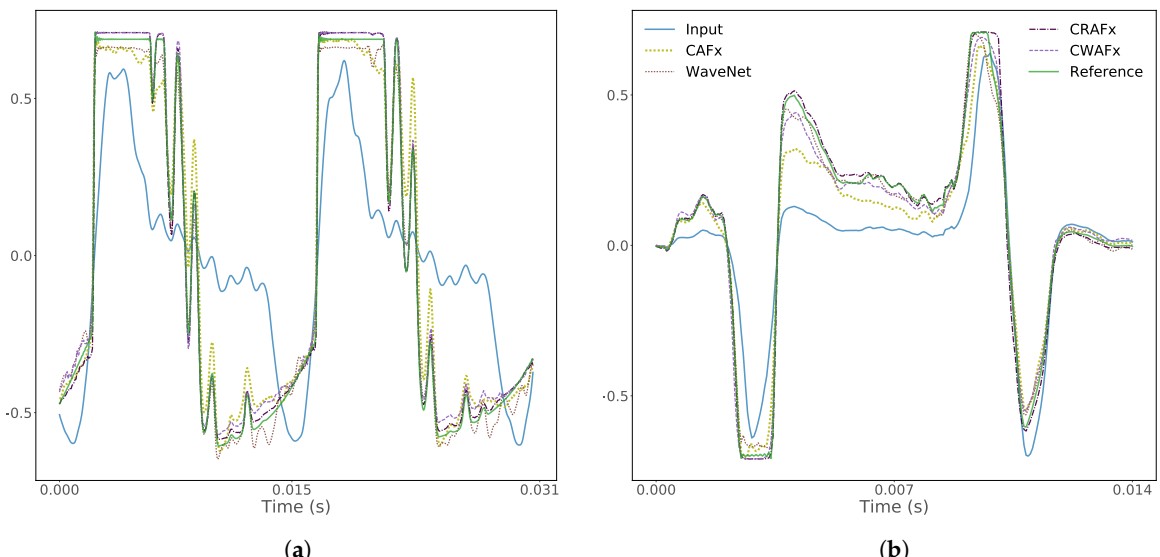

(**a**)　　　　　　　　　　　　　　　　　(**b**)

**Figure 8.** For the test samples from Figure 7, a segment of the respective waveforms: (**a**) **preamp** task and (**b**) **limiter** task when processing the onset of the input audio. Vertical axes represent amplitude.

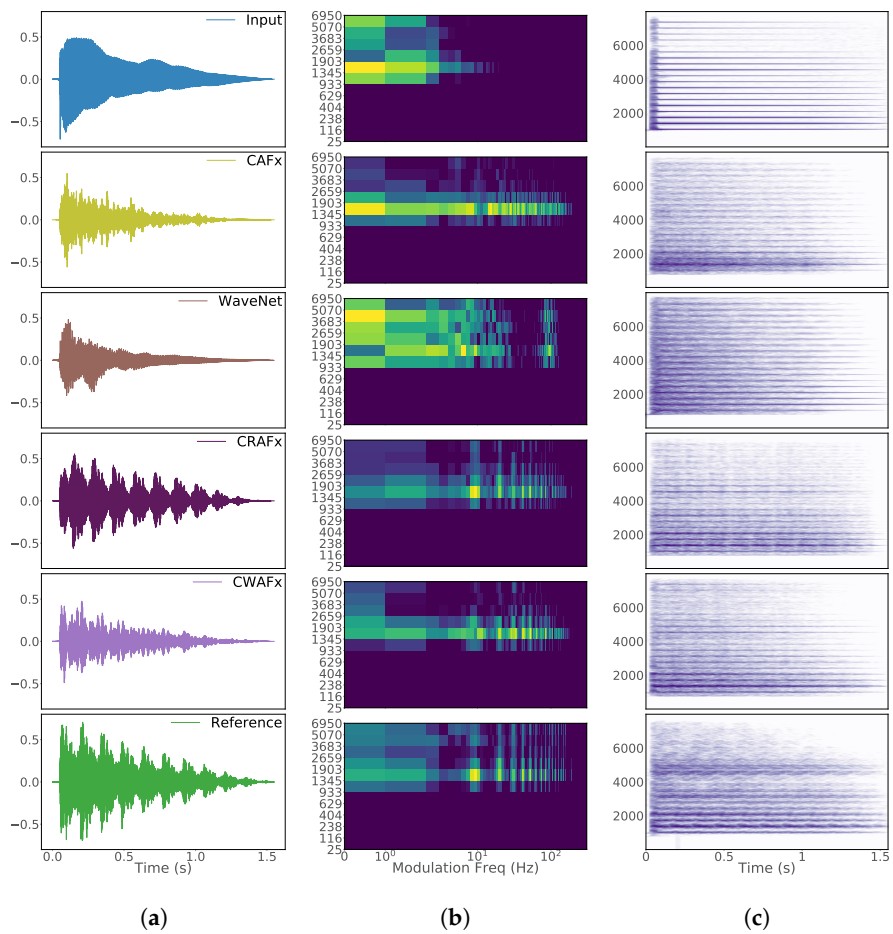

(**a**)　　　　　　　　　　(**b**)　　　　　　　　　　(**c**)

**Figure 9.** Results with selected samples from the test dataset for the **Leslie speaker horn-tremolo** tasks. (**a**) Waveform, (**b**) modulation spectrum and (**c**) spectrogram. Vertical axes represent amplitude, Gammatone frequency (Hz) and FFT frequency (Hz), respectively.

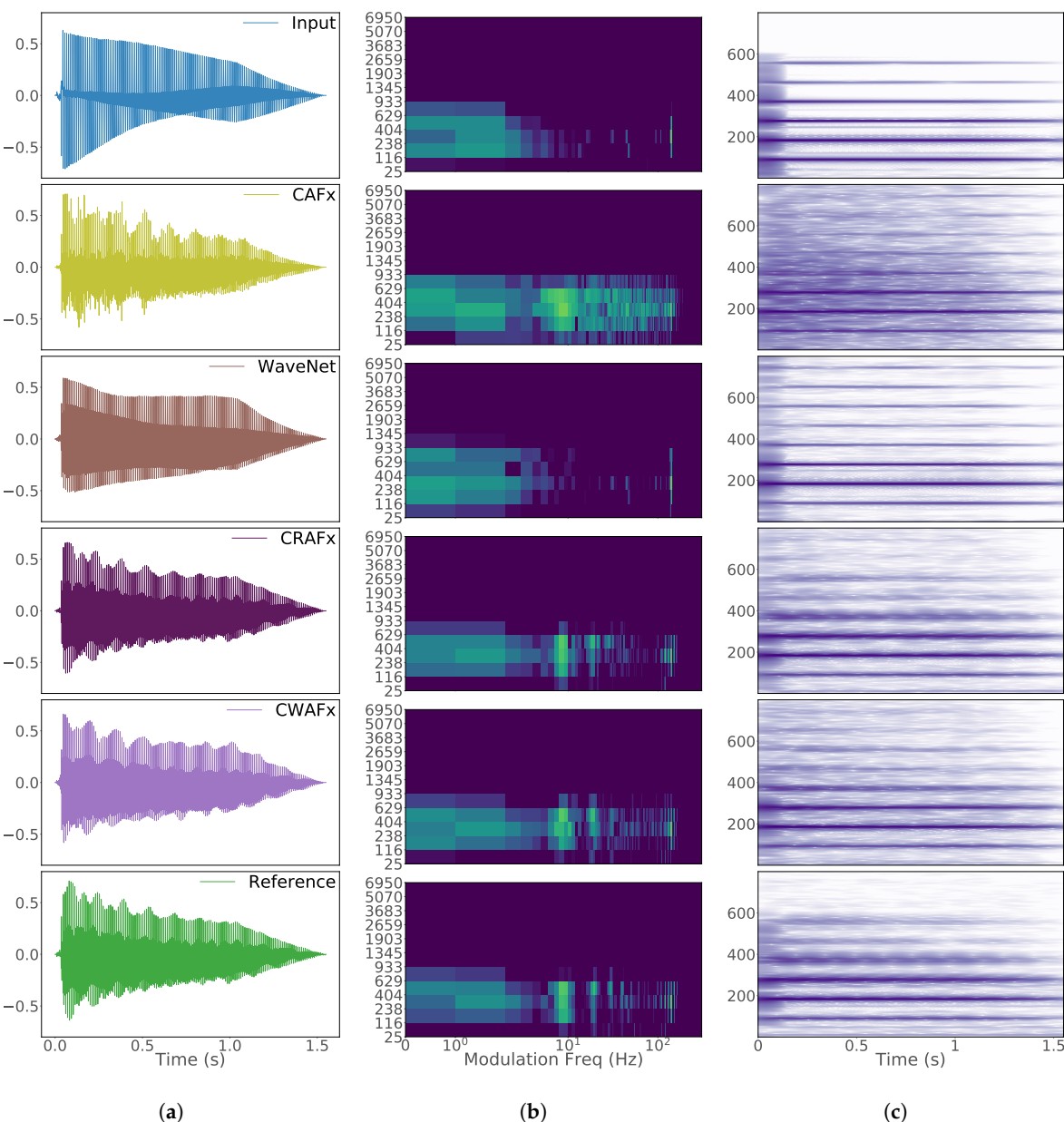

(**a**)              (**b**)              (**c**)

**Figure 10.** Results with selected samples from the test dataset for the **Leslie speaker woofer-tremolo** tasks. (**a**) Waveform, (**b**) modulation spectrum and (**c**) spectogram. Vertical axes represent amplitude, Gammatone frequency (Hz) and FFT frequency (Hz), respectively.

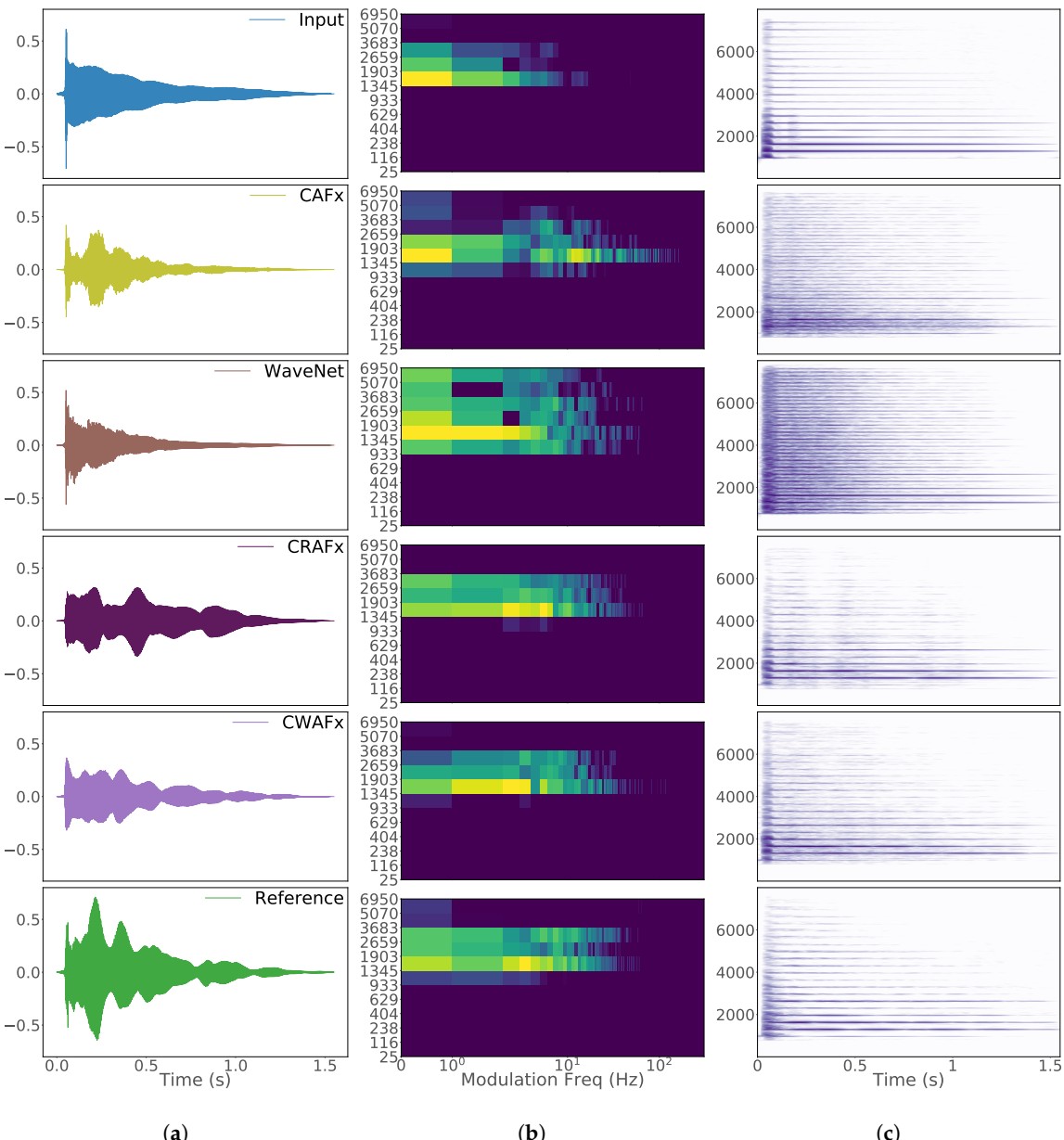

**Figure 11.** Results with selected samples from the test dataset for the **Leslie speaker horn-chorale** tasks. (**a**) Waveform, (**b**) modulation spectrum and (**c**) spectogram. Vertical axes represent amplitude, Gammatone frequency (Hz) and FFT frequency (Hz), respectively.

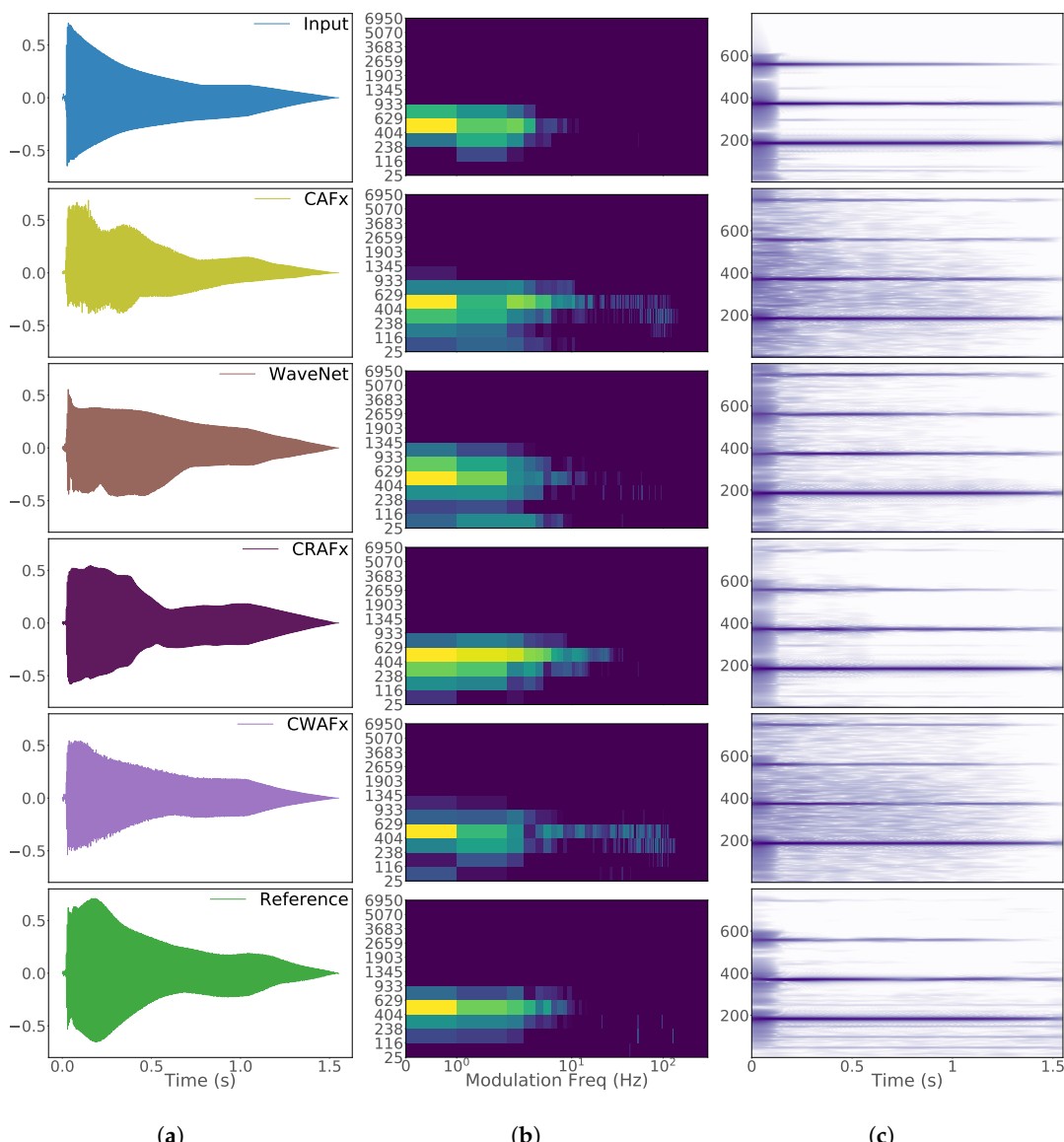

**Figure 12.** Results with selected samples from the test dataset for the **Leslie speaker woofer-chorale** tasks. (**a**) Waveform, (**b**) modulation spectrum and (**c**) spectogram. Vertical axes represent amplitude, Gammatone frequency (Hz) and FFT frequency (Hz), respectively.

## 6. Discussion

### 6.1. Nonlinear Task with Short-Term Memory - Preamp

The architectures that were designed to model nonlinear effects with short-term memory, such as *CAFx* and *WaveNet*, were outperformed by the models that incorporate temporal dependencies. With *CRAFx* and *CWAFx* being the highest scoring model both objectively and perceptually. Although this task does not require a long-term memory, the context input frames and latent-space recurrent and WaveNet layers from *CRAFx* and *CWAFx*, respectively, benefited the modeling of the *preamp*. This performance improvement could be on account of the temporal behaviour present on the vaccum-tube amplifier, such as hysteresis or attack and release timings, although additional tests on the *preamp* might be required.

Given the successful results reported in [7] and [9], which represent the state-of-the-art for nonlinear audio effects modeling, it is remarkable that the performance of these architectures (*CAFx*

and *WaveNet*) is exceeded by *CRAFx* and *CWAFx*. It is worth noting that the [7] model is trained with input frame sizes of 1024 samples, which could indicate a decrease in modeling capabilities when handling larger input frame sizes, such as 4096 samples. Similarly, the model from [9] included 1 stack of dilated convolutions whereas the *WaveNet* architecture used 2.

Nevertheless, from Figure 6a, we can conclude that all models successfully accomplished the modeling of the *preamp*. Most of the output audio is only slightly discernible from their target counterparts, with *CRAFx* and *CWAFx* being virtually indistinguishable form the real analog device.

### 6.2. Time-Dependent Nonlinear Task - Limiter

Since the *limiter* task includes long temporal dependencies such as a 1100 *ms* release gate, as expected, the architectures that include memory achieved a higher performance both objectively and subjectively. From Figure 7d it can be seen that *CAFx* and *WaveNet* introduce high frequency information that is not present in the reference spectrogram. This could be an indication that the models compensate for their limitations when modeling information beyond one input frame, such as the distortion tone characteristic due to the long release time together with the variable ratio of the *limiter*. Furthermore, from Figure 8b it is noticeable how each architecture models the attack behavior of the *limiter*.

We can conclude that although all networks closely matched the reference target, it is *CRAFx* and *CWAFx* which achieved the exact saturation waveshaping characteristic of the audio processor. The latter is accentuated with the perceptual results from Figure 6b, where *CRAFx* and *CWAFx* are again virtually indistinguishable from the reference target. While *CAFx* and *WaveNet* are ranked behind due to the lack of long-term memory capabilities, it is noteworthy that these models closely accomplished the desired waveform.

### 6.3. Time-Varying Task: Leslie Speaker

With respect to the *horn tremolo* and *woofer tremolo* modeling tasks, it can be seen that for both rotating speakers, *CRAFx* and *CWAFx* are rated highly whereas *CAFx* and *WaveNet* fail to accomplish these tasks. Thus, the perceptual findings from Figure 6c,d confirm the results obtained with the *ms_mse* metric and overall, the *woofer* task has a better matching that the *horn* task. Nevertheless, for *CRAFx* and *CWAFx*, the objective and subjective ratings for the *horn tremolo* task do not represent a significant decrease of performance and it can be concluded that both time-varying tasks were successfully modeled by these architectures.

*CRAFx* is perceptually ranked slightly higher than *CWAFx*. This indicates a closer matching of the reference amplitude and frequency modulations, which can be seen in the respective modulation spectra and spectrograms from Figure 9 and Figure 10.

For the *horn chorale* and *woofer chorale* modeling tasks, *CRAFx* and *CWAFx* successfully modeled the former while only *CRAFx* accomplished the *woofer chorale* task. Since the *woofer chorale* task corresponds to modulations lower than 0.8 Hz, we can conclude that Bi-LSTMs are more adequate than a latent-space WaveNet when modeling such low-frequency modulations.

In general, from Figure 9 to Figure 12, it is observable that the output waveforms do not match the waveforms of the references. This shows that the models are not overfitting to the waveforms of the training data and that the successful models are learning to introduce the respective amplitude and frequency modulations. The models cannot replicate the exact reference waveform since the phase of the rotating speakers varies across the whole dataset. For this reason, the early stopping and model selection procedures of these tasks were based on the training loss rather than the validation loss. This is also the reason of the high *mae* scores across the *Leslie speaker* modeling tasks, due to these models applying the modulations yet without exactly matching their phase in the target data. Further exploration of a phase-invariant cost function could improve the performance of the different architectures.

*CAFx* and *WaveNet* were not able to accomplish these time-varying tasks. It is worth noting that both architectures try to compensate for long-term memory limitations with different strategies. It is suggested that *CAFx* wrongly introduces several amplitude modulations, whereas *WaveNet* tries to average the waveform envelope of the reference. This results in output audio significantly different from the reference, with *WaveNet* being perceptually rated as the lowest for the *horn tremolo* and *horn chorale* tasks. This also explains the *ms_mse* results from Figure 5 for the *woofer chorale* task, where *WaveNet* achieves the best score since averaging the target waveform could be introducing the low-frequency amplitude modulations present in the reference audio.

## 7. Conclusions

In this work, we explored different deep learning architectures for black-box modeling of audio effects. Using raw audio and a given audio effects modeling task, we explored the capabilities of end-to-end DNNs to process the audio accordingly. We tested the models when modeling nonlinear effects with short-term and long-term memory such as a tube *preamp* and a transistor-based *limiter*; and nonlinear time-varying processors such as the rotating *horn* and *woofer* of a *Leslie speaker* cabinet.

Through objective perceptual-based metrics and subjective listening tests we found that across all modeling tasks, the architectures that incorporate Bi-LSTMs or, to a lesser extent, latent-space dilated convolutions to explicitly learn long temporal dependencies, outperform the rest of the models. With these architectures we obtain results that are virtually indistinguishable from the analog reference processors. Also, state-of-the-art DNN architectures for modeling nonlinear effects with short-term memory perform similarly when matching the *preamp* task and considerably approximate the *limiter* task, but fail when modeling the time-varying *Leslie speaker* tasks.

The nonlinear amplifier, rotating speakers and wooden cabinet from the *Leslie speaker* were successfully modeled. Nevertheless, the crossover filter was bypassed in the modeling tasks since the dry and wet audio were filtered accordingly. This was due to the limited frequency bandwidth of the bass and guitar samples, thus, this modeling task could be further explored with a more appropriate dataset such as Hammond organ recordings.

As future work, a cost function based on both time and frequency can be used to further improve the modeling capabilities of the models. In addition, since the highest ranked architectures use past and subsequent context input frames, more research is needed on how to adapt these architectures to overcome this latency. Thus, real-time applications would benefit significantly from the exploration of end-to-end DNNs that include long-term memory without resorting to large input frame sizes and the need for past and future context frames. Also, an end-to-end WaveNet architecture with a receptive field as large as the context input frames from *CRAFx* and *CWAFx* could also be explored for the time-varying modeling tasks.

Modeling of artificial reverberators such as a plate or spring can also be explored. Moreover, as shown in [9], the introduction of controls as a conditioning input to the networks can be investigated, since the models are currently learning a static representation of the audio effect. Finally, applications beyond virtual analog can be investigated, for example, in the field of automatic mixing the models could be trained to learn a generalization from mixing practices.

**Author Contributions:** Conceptualization, M.A.M.R., E.B. and J.D.R.; Data curation, M.A.M.R.; Formal analysis, M.A.M.R.; Investigation, M.A.M.R.; Methodology, M.A.M.R.; Supervision, E.B. and J.D.R.; Validation, M.A.M.R.; Visualization, M.A.M.R.; Writing—original draft, M.A.M.R.; Writing—review & editing, E.B. and J.D.R. All authors have read and agreed to the published version of the manuscript.

**Funding:** Emmanouil Benetos is supported by RAEng Research Fellowship RF/128. Joshua D. Reiss is funded by the EPSRC Programme Grant EP/L019981/1, 2014-2019.

**Acknowledgments:** The Titan Xp GPU used for this research was donated by the NVIDIA Corporation. The Queen Mary Ethics of Research Committee approved the listening test with reference number QMREC2165. The *Leslie speaker* samples were recorded with the help of Giulio Moro.

**Conflicts of Interest:** The authors declare no conflict of interest. The funders had no role in the design of the study; in the collection, analyses, or interpretation of data; in the writing of the manuscript, or in the decision to publish the results.

## Appendix A

Table A1 shows the number of trainable parameters and processing times across all the models. The latter was calculated for a *Titan XP GPU* and an *Intel Xeon E5-2620* CPU and corresponds to the time the model takes to process one batch, i.e., the total number of frames within a 2 s audio sample. GPU and CPU times are reported using the non real-time optimized *python* implementation.

**Table A1.** Number of parameters and processing times across various models.

| Model | Number of Parameters | GPU Time (s) | CPU Time (s) |
|---|---|---|---|
| *CAFx* | 604,545 | 0.0842 | 1.2939 |
| *WaveNet* | 1,707,585 | 0.0508 | 1.0233 |
| *CRAFx* | 275,073 | 0.4066 | 2.8706 |
| *CWAFx* | 205,057 | 0.0724 | 2.9552 |

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
