# Peer review of "Deep Learning for Black-Box Modeling of Audio Effects"

_applsci, doi:10.3390/app10020638_

Round 1

Reviewer 1 Report

General comments:

This paper gives a small review of work in the are of modelling audio effects with machine-learning techniques, and then applies a selection of generalized methods (both pre-exisisting and novel) to modelling a selection of audio effects. The goal of the author seems to be to derive a network that can model both effects with short time-dependencies, and ones with very long time dependencies (for example LFO modulation).

The paper is generally well written and constructed, but there's some problems that need to be addressed before publication:

- The discussion of virtual analog topics in general needs some review by the authors, there's some misuse of terminology and concepts which is quite misleading.
- The new CRAFx and CWafx architechures are well described in terms of the structure of the algorithms, but poorly justified. I'm really missing some discussion about why decisions about the architecture were made, and for what purpose. At the moment, the presentation suffers from feeling rather arbitrary. This then lessens the value of the results, as its hard to draw a conclusion about why these networks perform better than previous ones.
- There's no discussion about computational complexity. This is crucial in a subject area where real-time processing is overwhelmingly dominant.

Specific notes:

{L.12-14, L.67-74, Table 1, L.156}
There seems to be some confusion with the authors about the term 'memoryless'. Any system (analog or digital) that contains state has memory by definition. Distortion circuits, amplifiers, tubes etc are absolutely not memoryless. Neither are the methods used to model them (beyond a simple mapping function, which is rarely used). The authors should not state this. The intention seems to be draw a distinction between a circuit like a distortion that usually has a short memory and one like a compressor that has a longer memory, but this is not the standard meaning of the term.

{L.97, Ref 12.}
This paper does not describe a memoryless static waveshaper.

{L.102 -L.103}
The implication that Volterra and Wiener-Hammerstein techniques somehow linearize a nonlinear system is not correct. They are nonlinear models of nonlinear systems.

{L.180}
What does hop size mean in this context?

{L.206}
Given that a dense network of sufficient capacity and a standard activation function is already a universal approximator for nonlinear functions, what is the motivation for using SAAF in addition?

{Table 4}
The number of params reported for the LSTM seems to be too low? Maybe I'm misunderstanding?

{L.309 - 316}
I'd appreciate some perspective on the number of epochs needed for training here. Is it hundreds, thousands, tens of thousands?

Reviewer 2 Report

Review of “Deep Learning for Black-Box Modeling of Audio Effects” by Ramirez, Benetos and Reiss

The authors of this manuscript present various machine learning approaches to model analog audio effects that are known to contain nonlinearities. By systematic comparison between the various architectures, they find that models containing recurrent connections, as expected, are able to learn long term dependencies in the data and therefore outperform other architectures.

Barring a few typographical and grammatical errors, the manuscript is well written and the data, modeling and error analysis are presented well. I especially appreciate the fact that the authors did a through quantitative comparison of various models that clearly presents the benefits of the recurrent models over the DNNs.

While not necessarily to this analysis, I would be curious to know the number of parameters across various models.

Nevertheless, the manuscript is acceptable for publication in Applied Sciences.

Author Response

While not necessarily to this analysis, I would be curious to know the number of parameters across various models.

To provide this information, Appendix A was added.